# Long-range movement of large mechanically interlocked DNA nanostructures

Jonathan List[1], Elisabeth Falgenhauer[1], Enzo Kopperger[1], Günther Pardatscher[1] & Friedrich C. Simmel[1]

Interlocked molecules such as catenanes and rotaxanes, connected only via mechanical bonds have the ability to perform large-scale sliding and rotational movements, making them attractive components for the construction of artificial molecular machines and motors. We here demonstrate the realization of large, rigid rotaxane structures composed of DNA origami subunits. The structures can be easily modified to carry a molecular cargo or nanoparticles. By using multiple axle modules, rotaxane constructs are realized with axle lengths of up to 355 nm and a fuel/anti-fuel mechanism is employed to switch the rotaxanes between a mobile and a fixed state. We also create extended pseudo-rotaxanes, in which origami rings can slide along supramolecular DNA filaments over several hundreds of nanometres. The rings can be actively moved and tracked using atomic force microscopy.

[1] Physik-Department E14, Technische Universität München, Am Coulombwall 4a, 85748 Garching, Germany. Correspondence and requests for materials should be addressed to F.C.S. (email: simmel@tum.de).

Over the past decades, supramolecular chemists have developed efficient synthesis procedures for the generation of mechanically interlocked molecules[1–4], and potential applications for such structures are emerging. For instance, rotaxanes have already been used as force-generating components for molecular elevators[5], molecular pumps[6], as switches in molecular electronics[7], and for the control of stepwise chemical synthesis[8].

DNA nanotechnology represents an alternative strategy for the generation of supramolecular structures that utilizes sequence-programmable interactions between DNA molecules. DNA nanostructures can be produced by automated design and synthesis procedures, and thus enable a fast exploration of supramolecular designs. Already in the 1990s, a variety of DNA-based knots and interlocked rings[9,10] were synthesized, followed later by the realization of switchable catenane structures[11,12] and origami catenanes produced by molecular kirigami[13]. In 2010, the first DNA rotaxane structures were created[14], which comprised a linear double-stranded DNA axis and circular stoppers. Later mechanically more rigid[15], light-switchable[16] and Daisy chain[17] DNA rotaxanes were realized, and also rotaxanes involving gold nanoparticles[18]. For a recent review see ref. 19.

Most of the work on interlocked DNA structures used single-stranded or double-stranded DNA as a building material. Thus, even though the structures were topologically well defined, they were mechanically rather flexible. By contrast, DNA origami structures[20,21] consist of several, multiply connected DNA double helices in parallel and thus exhibit a much larger rigidity. Until now, however, such structures were mainly used as rigid scaffolds, but not for the realization of molecular machinery. Notable exceptions were the production of switchable DNA nano-containers[22,23] or the realization of DNA-based kinematic pairs, which were inspired by macroscopic engineering mechanisms[24,25]. In these examples, the underlying DNA origami structures were folded from a single scaffold strand and thus consisted of a single subunit. As a consequence, the structures were only capable of restricted local movements.

We here demonstrate the construction of large and structurally rigid DNA origami rotaxane structures from multiple subunits—origami macrocycles threaded onto rigid multi-helix axles with bulky stoppers. Origami pseudo-rotaxanes, in which the macro-cycles move along elongated axles over several 100 nm, indicate the potential of this approach for the creation of mesoscopic molecular machines displaying processive long-range motion. As a consequence of the purely mechanical bond, origami rotaxanes are promising functional components for the creation of molecular transporters, enabling fast and directionally guided sliding mobility over long distances.

## Results

### Formation of two-component rotaxane constructs.

Rotaxanes were synthesized using a clipping approach[26] based on two subunits, which were each created using the DNA origami technique[21] (Fig. 1a). One subunit—the rotaxane axle—was designed in the shape of a dumbbell, comprised of a linear axis and stopper elements at each end. To facilitate efficient assembly of an interlocked structure, the rotaxane ring subunit was first produced separately in a flexible, open configuration and localized at the dumbbell axis via sticky end hybridization (for transmission electron microscopy (TEM) images of the subunits see Supplementary Figs 1–8. A detailed description of the design see the Supplementary Methods and Supplementary Figs 24–27. For a list of DNA sequences used see Supplementary Data 1). Closing strands were then added to join the ring around

the axle. Subsequently, a set of fuel strands served to remove the temporary connection between ring and axle via toehold-mediated strand displacement[27], resulting in a fully detached ring sliding on the axle between the stoppers.

We applied this general approach to two alternative designs. The first rotaxane (termed R1D1, Fig. 1b,c and Supplementary Figs 9–10) with overall dimensions of $\sim 140 \times 40$ nm was constructed from a dumbbell module (D1) and a tubular ring (R1). The dumbbell was composed of a 10-helix axis placed between bulky stopper blocks at both ends. Nine axis staple strands were extended to act as sticky ends for the temporary attachment of the macrocycle. The open ring R1 was composed of two separate rigid halves connected by a flexible hinge, of which one carried nine sticky ends for attachment to the dumbbell. Closure of the hinged structure resulted in a hexagonal tubular toroid consisting of 76 parallel helices with a total length of 35 nm and an inner diameter of 15 nm (cf. Supplementary Table 1 for all dimensions).

An alternative design (termed R2D2, Fig. 1d–f and Supplementary Figs 11–12) was implemented using curved structural elements as introduced in ref. 28. An H-shaped structure with dimensions $90 \times 71$ nm was created to serve as the rotaxane axle. The rotaxane ring consisted of 14 double helices bent into a torus with an inner diameter of 28 nm. An additional 28-helix block attached to the torus acted as an orientation marker. As above, the ring was folded into an open configuration, which contained a gap large enough for the axle to pass through (Fig. 1d). After its localization to the axle, closing strands were added that forced the ring into a closed torus conformation (Fig. 1e).

Quality control by TEM indicated reasonable yields for the localization of the ring for both rotaxane designs (R1D1 85%, R2D2 87%; see Supplementary Methods) even without final gel purification.

### Switching between mobile and immobile state.

To assess whether the rings were indeed free to move along the axles, we investigated TEM micrographs taken before and after the release of the rings in greater detail (Fig. 2). The translational movement of rotaxane R1D1 was analysed by measuring the distance of ring R1 from its nominal initial position on the axle. Only a translational small shift between the initial position and the average ring position could be expected if all rings were successfully released, as the attachment position was designed to bind the ring only 3,9 nm from the centre of the axle. The resulting bimodal distributions for the ring position (Fig. 2b) indicate that the ring either resides on the initial attachment point, or slips off and assumes a distal position 15 nm away from it. This suggests that after release of the ring the attachment staples constitute a sterical barrier for the ring, preventing it from sliding back and evenly distribute along the axle (cf. Supplementary Fig. 25a,b for a cross-section of R1D1). Most importantly, the fraction of rings found on the distal position significantly increases after the addition of release strands (Fig. 2d).

The rotational state of R1 with respect to the D1 axle could not be visualized in these images. We therefore attached gold nanoparticles (AuNPs, 10 nm diameter) as labels to both ring and axle, which could be clearly identified in the TEM images (Fig. 2c, see also Fig. 3a). Initially, AuNPs were bound to the unreleased rotaxanes in a *cis* configuration. After release, a larger fraction of particles was found in *trans* configuration, indicating a rotational movement of the ring (Fig. 2e). A finite fraction of rings apparently had slipped off the attachment site even before the addition of release strands. This may be caused by the ring closure process itself, which creates a crowded and potentially strained state at the attachment site. Furthermore, deformation of the structures during adhesion to the TEM grid or the staining

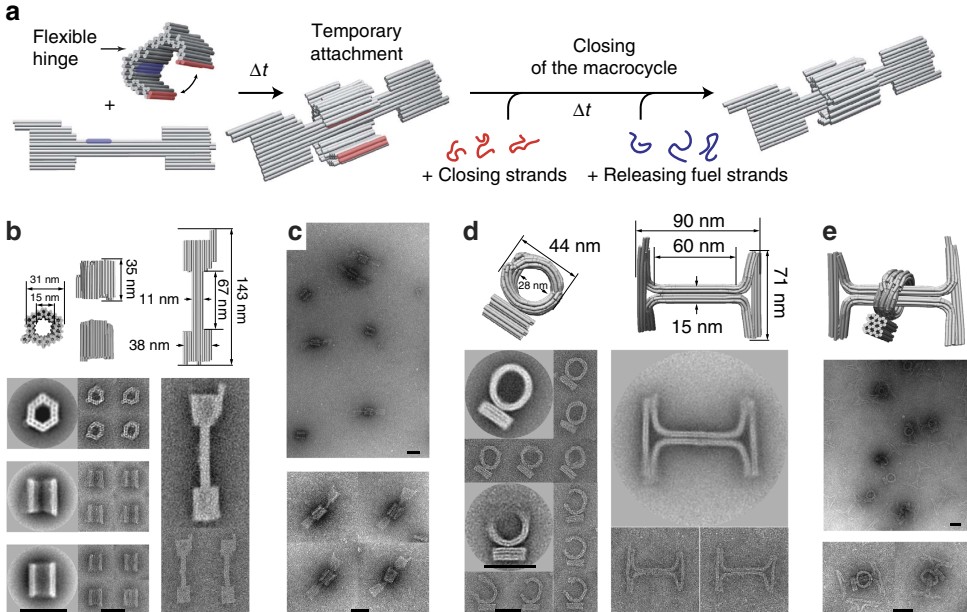

**Figure 1 | Fabrication of DNA origami rotaxanes.** (**a**) A rotaxane is formed from an open ring (R1) with a flexible hinge and a dumbbell-shaped DNA origami structure (D1), which were prepared separately. The hinge of the ring consists of a series of strand crossovers into which additional thymines are inserted to provide higher flexibility. Ring and axis subunits are first connected and positioned with respect to each other using 18 nucleotide long, complementary sticky ends 33 nm away from the centre of the axis (blue regions). The ring is then closed around the dumbbell axis using closing strands (red), followed by the addition of release strands that separate dumbbell from ring via toehold-mediated strand displacement. (**b**) 3D models and corresponding averaged TEM images of the ring and dumbbell structure. Also shown are exemplary single-particle images. (**c**) TEM images of the completely assembled rotaxanes (R1D1). (**d**) 3D models, averaged and single-particle TEM images of R2 and D2, subunits of an alternative rotaxane design containing bent structural elements. The TEM images of the ring structure correspond to the closed (top) and open (bottom) configurations. (**e**) 3D representation and TEM images of the fully assembled R2D2 rotaxane. Origami models are generated using CanDo. Scale bar, 50 nm. See also Supplementary Figs 1–18 for additional TEM images, and Supplementary Fig. 19A for an AFM image of R1D1 rotaxanes. 3D, three dimensional.

and drying processes may rip some of the rings out of their nominal starting position.

For the R2D2 design, we observed a much clearer ring movement after addition of the release strands. To determine the efficiency of the release process in this case, we counted the fraction of rotaxanes in one of two distinct positional states (centred and distal), which were observable in the TEM images (Fig. 2f). After addition of the release strands, an almost 10-fold increase of the distal population was observed (Fig. 2g). Judging from the orientation of the marker block on R2, we also observed rotational mobility of the ring (Fig. 2h).

In the following, we focused on the R1D1 design for further characterization. To demonstrate the potential use of the rotaxanes for the transport of nanoscale objects, we functionalized the rotaxane also with 20 and 30 nm diameter gold particles (Fig. 3a). Again, TEM images showed a clear lateral and radial displacement of the structures upon release of the rings. We further studied the mobility of the rotaxane using bulk fluorescence resonance energy transfer (FRET) measurements, for which R1 and D1 were labelled with four FRET donor–acceptor pairs close to the initial attachment position (Fig. 3b). Using a toehold-mediated strand displacement mechanism[29], we repeatedly switched the R1D1 rotaxane between a localized state, in which the ring was attached to its initial position via adaptor strands, and a mobile state (Fig. 3c). The connection between ring and dumbbell was broken by the addition of a set of fuel strands binding to the ring's adaptors. To return the ring to its initial position, anti-fuel strands were added to remove the fuel strands, reactivating the adaptors for binding. Direct binding and blocking of the dumbbell attachment sites by the anti-fuel strands was avoided by using two separate anti-fuels per adaptor strand,

which each were only partially complementary to the dumbbell adaptors. Using high fuel concentrations, the release and rebinding processes occurred within minutes. For instance, the final release step shown in Fig. 3c had a half-time of only about 40 s when using fuel strands at a concentration of 640 nM. In addition to bulk FRET measurements, we also performed super-resolution microscopy experiments using the DNA-PAINT technique[30], demonstrating a small mean distance change between ring and stopper elements consistent with the translational movement of the ring (Supplementary Figs 22 and 23).

**Extended multi-component rotaxane constructs**. To be able to visualize the ring mobility for the R1D1 rotaxane more clearly, we next created rotaxane axles with a much longer axis. To this end, we replaced D1 by a module that consisted of a stopper with 94 and 15 nm long axle sections on its both sides and an attachment site for R1 on the 94 nm axle (Fig. 4a). Rings were assembled on individual stopper modules, followed by polymerization of the R1-stopper complexes via dedicated polymerization staples. As shown in Fig. 4b, this resulted in elongated chains with multiple rotaxane structures in series, on which some of the rings were clearly mobile. We further extended the rotaxane axle by inserting a 246-nm-long DNA 10-helix bundle between two stopper modules, resulting in rotaxane molecules with a total axle length of 355 nm, on which the ring R1 was free to move (Fig. 4c).

**AFM manipulation of extended pseudorotaxane filaments**. Finally, we created even longer tracks for the R1 rings by polymerizing axle modules without stoppers into elongated pseudorotaxane filaments (Fig. 5a,b, see Supplementary

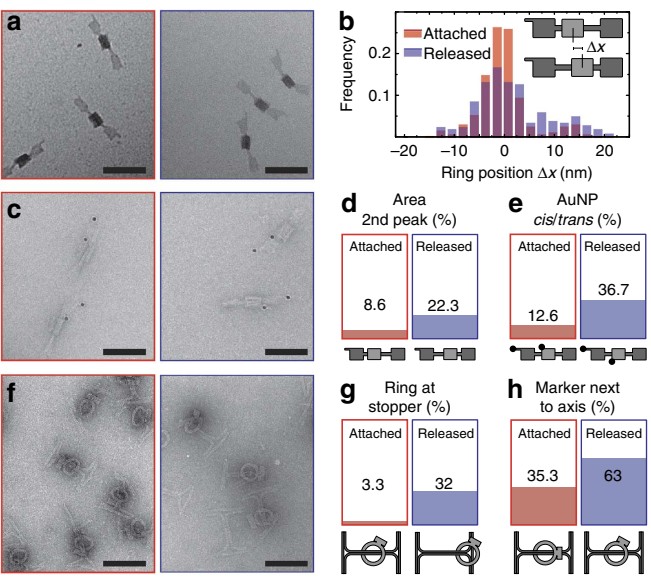

**Figure 2 | Investigation of ring mobility using TEM. (a)** Images of the R1D1 rotaxane before (left) and after (right) the addition of release strands. (**b**) Histogram of the distances of the rings from the mean initial attachment position before and after release. The distributions can be fitted by a sum of two Gaussians. A broadening of the main peak and a larger fraction of rings in the second peak is observed after release. (**c**) Images of R1D1 structures labelled with 10 nm gold nanoparticles, which serve as markers for highlighting the rotational orientation of the components relative to each other. Images were taken without (left) and with (right) release strands. (**d**) Ratio of particles found in the second peak of the Histogram shown in **b**. The rings displaced from their starting position indicate translational mobility of the construct. (**e**) Analysis of the rotation of R1D1. Gold particle markers were mostly found in their *cis* starting configuration before the release of the rings. After release, a considerably larger fraction is found in the *trans* position. (**f**) R2D2 TEM images before (left) and after (right) release. (**g**) Particles are classified into two states—one in which R2 is in the middle of the axis, and another in which R2 sits on top of the stopper bars. After release of the rings, an increase of the fraction of rings in the remote ring position is observed. (**h**) Rotational movement of the rings was detected by comparing the number of particles with the marker block lying above or under the axis (initial attachment position) with those where the marker is found off-axis. Upon release the radial orientation of about 30% of all rings changed. Scale bar, 100 nm.

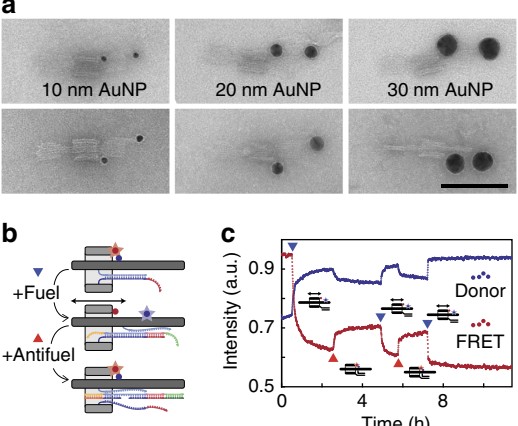

**Figure 3 | AuNP functionalization and bulk fluorescence experiments.** (**a**) R1D1 rotaxanes were modified with 10, 20 and 30 nm gold nanoparticles (from left to right), demonstrating the potential use of the rotaxanes as functional nanomechanical devices. Rotaxanes before addition of the release strands are shown in the first row, after addition of the release strand set in the second row. Scale bar, 100 nm. See also Supplementary Figs 15–19 for additional TEM and AFM images. (**b**) Scheme of the fuel/anitfuel mechanism used to switch the connection between R1 and D1. (**c**) Bulk spectroscopy experiments with R1D1 rotaxanes labelled with Cy3/Cy5 FRET pairs. Fuel and anti-fuel strands were added repeatedly (marked as blue and red triangles) with increasing concentrations (40 nM fuel, 40 nM anti-fuel, 160 nM fuel, 160 nM anti-fuel and 640 nM fuel). A decrease in the intensity of the FRET signal was observed upon release, as well as a recovery after the reattachment of the ring due to the displacement of the release strands. A moving average filter was applied and the intensities were normalized using the acceptor signal.

Figs 13–18 for additional TEM images of multi-component structures). Using fast scanning atomic force microscopy (AFM), we directly observed the mobility of the macrocycles along the axle (Fig. 5c,d, Supplementary Fig. 20 and Supplementary Movie 1). In these experiments, the motion of the rings was in fact driven by the AFM tip and the rings sled along the filaments parallel to the slow scanning axis. The tip-induced movement could only be observed for filaments that were weakly fixed to the mica substrate, which made imagining particularly challenging. Because of the adsorption of filaments and rings onto the mica substrate, however, it was not possible to observe thermally driven, diffusive motion of the rings. In future experiments, rotaxanes could be attached to elevated posts[31], which would enable observation of the free gliding of the rings.

## Discussion

Using the DNA origami technique, we have demonstrated the fabrication of biomolecular rotaxane structures with axle lengths up to 355 nm and pseudo-rotaxanes extending even in the micrometre range. Compared with earlier approaches, the resulting structures were much larger and structurally more rigid. Successful assembly and mobility of the structures were demonstrated using various characterization techniques, including electron microscopy and AFM. Reversible switching between the localized and mobile ring configuration was shown using bulk FRET experiments. Sliding of a single ring along an origami filament over several hundred nanometres could be demonstrated in AFM experiments, in which the ring was actively pushed along its track. We also functionalized rotaxane structures with gold nanoparticle cargoes of various sizes, indicating their potential use as transporters for nanoscale objects.

Long-range transport using DNA origami rotaxane structures is fundamentally different from previous approaches based on DNA-based walkers[32–35]. The interlocked nature of rotaxanes ensures localization of the mobile ring at the track without requiring a tight bond between them. While the speed of molecular walkers is ultimately limited by the timescales required for binding to the track or unbinding from it (potentially including additional enzymatic steps and conformational changes), the origami rings could slide from one binding site to a distant site simply by diffusion—which in principle can be much faster. In order to enable free sliding of the rings, the rotaxane structures would have to be elevated from the substrate, or operated in solution or in a gel matrix. Even though diffusive ring sliding is non-directional, transport of components from one docking site to another could be specifically controlled through the sequence of the adaptor strands, acting as unique localization addresses. Transport could actually be rendered directional by modulating the ring-binding strengths along the tracks to create a

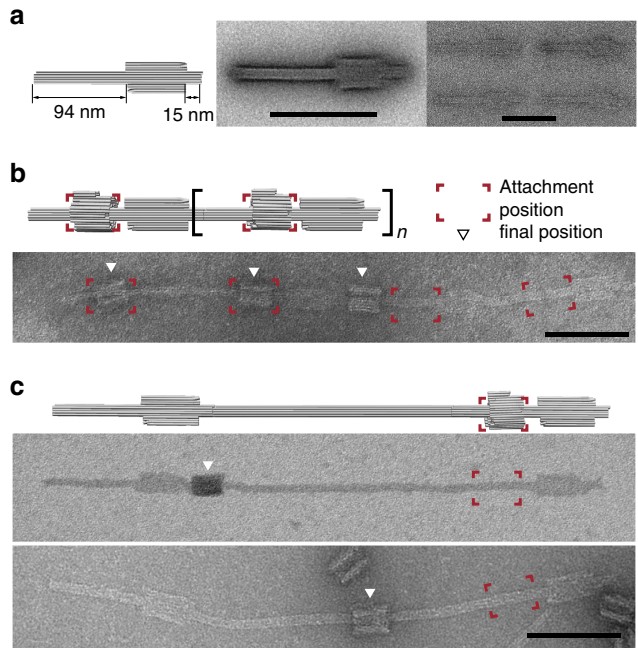

**Figure 4 | Elongated rotaxane structures composed of multiple components.** (**a**) 3D model, averaged and single TEM images of a 'stopper module' (rotaxane axis with integrated stopper). (**b**) Schematic representation of a rotaxane chain created by polymerization of stopper modules with attached macrocycles. The corresponding TEM image shows the chain after the release of the macrocycles. The initial attachment positions are highlighted in the red frames. (**c**) Extended rotaxane constructed from two stopper modules separated by a 246-nm-long axis module. The ring was initially attached close to one of the bumper pieces. TEM images taken after the release of the rings demonstrate sliding mobility along the axis. Scale bars, 100 nm. 3D, three dimensional.

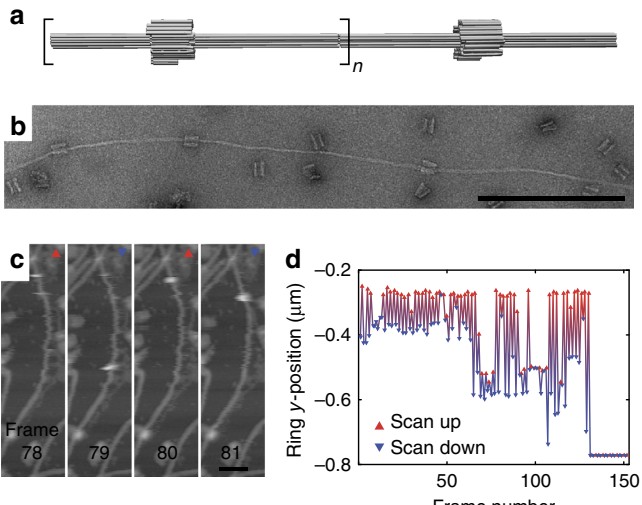

**Figure 5 | Pseudorotaxane filaments with multiple rings.** (**a**) 3D model of filaments composed of polymerized axis modules with R1 rings attached. (**b**) TEM image of a pseudorotaxane filament. Rings are still attached at their starting position. Scale bar, 500 nm. (**c**) Fast scan AFM (scan rate 19.5 Hz, 0.076 frames s$^{-1}$) snapshots of a pseudorotaxane electrostatically immobilized on the mica substrate (the full image sequence is available as Supplementary Movie 1). The macrocycle is pushed along the filament by the AFM tip in the slow scanning direction. An overview AFM image of a pseudorotaxane sample is shown in Supplementary Fig. 19B. Scale bar, 100 nm. (**d**) Y-position or the ring for all images shown in Supplementary Movie 1. The slow scan direction is indicated by the triangles. 3D, three dimensional.

rotaxane structures by gel electrophoresis see Supplementary Fig. 21. CaDNAno design maps for the structures are displayed in Supplementary Figs 28–31.

ratchet potential[36], which could be switched by plasmonic thermal cycling[37] or optical switching[38–41].

As one of the major advantages of DNA origami structures, both ring and axle of the rotaxanes can be easily modified with multiple cargo molecules and nanoparticles, at nanometre-precise locations and with defined stoichiometries. The potential to transport these components quickly over micrometre distances may find application in the realization of programmable assembly lines[42], sequential DNA-templated synthesis[43], and the control of DNA-directed chemical processes at a distance.

## Methods

**DNA origami structures.** The rotaxane subunits were produced by thermal annealing of a solution containing 50 nM origami scaffold (a 7,249 nucleotide (nt) long strand was used, except for ring1 and dumbbell D2, which were created from a 7,560 nt long scaffold[44]), 200 nM scaffold strands, 20 mM MgCl₂ and 1 × TAE buffer (40 mM Tris 20 mM acetic acid 1 mM EDTA). The structures were folded using 4 h long thermal annealing ramps over 4 °C (axle module: 60–57 °C; D1: 58–55 °C; R1, R2, D2: 56–53 °C; stopper module: 54–51 °C, see also Supplementary Table 2). Excess staples were then removed using PEG precipitation[45]. To this end, a 15% PEG 8,000 solution containing 505 mM NaCl, 20 mM MgCl₂ and 1 × TAE was added to the sample at an equal volume, followed by centrifugation for 30 min at 20,000 r.c.f. The supernatant was removed and the pellet was resuspended in a buffer containing 505 mM NaCl, 20 mM MgCl₂ and 1 × TAE and vortexed for at least 2 min. This procedure was repeated three times followed by resuspension of the pellet in the final step in one-fourth of the initial volume (1 M NaCl, 20 mM MgCl₂ and 1 × TAE) in order to obtain structures at a high concentration (200 nM). The resulting solution was shaken at 600 r.p.m. for at least 1 h at ≈ 37 °C to ensure that the structures were dissolved. Ring and dumbbell monomers were mixed in a 1:1 ratio and incubated for 1 week at 30 °C to create ring-dumbbell dimers. Subsequently the ring was closed by adding the corresponding staple subset and incubating the sample at 35 °C for several hours. For quality control of the

**Functionalization of origami structures with AuNPs.** Functionalization of the origami structures with gold nanoparticles was performed as previously described[46]. First the nanoparticles were concentrated and coated with DNA. To this end, 50 ml gold particles (BBI solutions) at the concentration supplied by the distributor (for example, 10 nM for 10 nm particles) were coated with BSPP Bis(p-sulfonatophenyl)-phenylphosphine dihydrate dipotassium salt) by adding 4 mg ml$^{-1}$ BSPP and constantly shaking at room temperature over 3 days to avoid aggregation during the functionalization process. A 5 M NaCl solution was added to the nanoparticle solution until the colour changed from red to blue and the particles were centrifuged at 1,600 r.c.f. for 30 min. The supernatant was removed and the pellet was dissolved again in 800 µl of 2.5 mM BSPP solution. Methanol (800 µl) was added followed by another centrifugation step of 1,600 r.c.f. for 30 min. The resulting pellet was dissolved in 800 µl of BSPP solution, the concentration was determined using an absorption spectrometer and the concentrated nanoparticles were coated with thiolated DNA (5′-HS-TCTCTCTCTCTCTCTCTCTC-3′). The thiolated DNA was treated with 10 mM TCEP (Tris (carboxyethyl) phosphine hydrochloride) for at least 30 min before adding to the nanoparticles. For 10 nm particles a 100-fold excess of oligonucleotides over the particles was used (400-fold for 20 nm and 900-fold excess for 30 nm particles). The coating process was accelerated at low pH conditions and therefore citrate buffer (pH 3) was added to a final concentration of 20 mM. After constant shaking for 1 h (the long duration was only necessary for large nanoparticles, the 10 nm AuNPs could be coated within 3 min) the pH-value was raised again by adding HEPES (pH 7.6) to a final concentration of 100 mM. The progress of coating process was tested by adding a magnesium chloride solution (125 mM) to a small amount of the sample. While uncoated particles aggregate at these conditions—accompanied by a colour change from red to blue—, the DNA coated particles stay in solution. Subsequently the remaining unbound thiol strands were removed by filtration with 0.5 × TBE buffer using 0.5 ml 100 kDa Amicon ultra-centrifugal filters (Merck Millipore). The samples were filtered at 8,000 r.c.f. for six minutes for five rounds. This procedure was afterwards repeated with a fresh filter. Purification was performed immediately before adding the AuNPs to the origami rotaxanes.

Rotaxane structures were modified with the AuNPs after assembly and closure of the rings. Particles were conjugated to the rotaxanes by mixing the particles (after addition of MgCl₂ to a final concentration of 20 mM and 1 × TAE buffer)

with the structures at fivefold excess per binding site and incubate them over night on a shaker. Each of the binding sites on the origami structures consists of three staple strands extended by the complementary sequence. After conjugation, excess gold particles were removed from the sample by gel electrophoresis (0.5% agarose gel, $1 \times$ TAE 12.5 mM MgCl$_2$). The origami band cut from the gel and the sample was extracted by squeezing. Centrifuging the product at 20,000 r.c.f. for 20 min and removing supernatant up to the target volume was used to increase the sample concentration to the desired value and the pellet was re-dissolved on a shaker at 37 °C for 30 min. In general, we found that simple centrifugation is a facile method to highly concentrate AuNP-modified origami structures, which could be useful also for applications in plasmonics. With this method, also a complete buffer exchange can be performed easily.

**AFM imaging.** AFM images were recorded using an Asylum Research Cypher AFM and Olympus BioLever mini cantilevers (spring constant 0.05–1.2 N m$^{-1}$). Typically a drive frequency of 18 kHz was used. Five microlitre of about 2 nM origami sample (20 mM MgCl$_2$ and $1 \times$ TAE buffer) were added onto freshly cleaved mica. As 10-helix bundles interact with the mica substrate only via a relatively small area, we added 30 μl of 125 mM MgCl$_2$ and $10 \times$ TAE buffer to sufficiently immobilize the structures on the substrate. Release strands were added in excess to disconnect the attached rings.

**Transmission electron microscopy.** Negative stain samples were prepared on glow-discharged formvar-supported carbon-coated Cu400 TEM grids (Science Services, Munich, Germany). A total of 5 μl of sample solution were adsorbed on the grid for 30 s and subsequently stained with 2% aqueous uranyl formate solution containing 25 mM NaOH for 40 s. Samples were then dried with filter paper. Images were recorded with a Philips CM100 transmission electron microscope at 100 kV and an AMT $4 \times 4$ Megapixel CCD camera.

**FRET experiments.** FRET experiments were carried out using a Cary Eclipse spectrometer (Agilent Technologies Deutschland GmbH, Böblingen, Germany). The donor dye (Cy3) was excited at $550 \pm 5$ nm and observed at $575 \pm 5$ nm, while the acceptor (Cy5) was excited at $650 \pm 5$ nm and observed at $675 \pm 5$ nm. The FRET signal was acquired at an emission of 675 nm when excited with 550 nm light. Samples were diluted to a concentration of 10 nM and for each experiment 65 μl of sample were filled into fluorescence cuvettes (105.254-QS, Hellma GmbH & Co. KG, Müllheim, Germany). Release and anti-release strands were added during data acquisition and the solution was mixed vigorously with a pipette. Data points were recorded every 6 s with an integration time of 1 s. Temperature was kept constant at 37 °C during the measurement.

**Data availability.** The data that support the findings of this study are available from the corresponding author upon request.

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

## Acknowledgements

We gratefully acknowledge financial support by the Deutsche Forschungsgemeinschaft (SFB 1032 TPA2 and cluster of excellence Nanosystems Initiative Munich (NIM)). G.P. is supported by the Volkswagen Stiftung (grant no. 86 395). We thank K. Kapsner for helpful discussions and F. Praetorius for providing DNA scaffold.

## Author contributions

J.L., E.F. and F.C.S. planned and designed the experiments; J.L., E.F., E.K. and G.P. performed the experiments; J.L., E.F., E.K. and F.C.S. wrote the paper; and all authors discussed the results and commented on the manuscript.

## Additional information

**Competing financial interests:** The authors declare no competing financial interests.

