## [Peer review file · Nature Communications]

REVIEWERS' COMMENTS:

Reviewer #1 (Remarks to the Author):

List et al. report construction of DNA origami rotaxanes based on the honeycomb lattice design. They employed two kinds of designs: a composite of a dumbbell module and a tubular ring, and of a curved axle origami and a torus ring. The former strategy was applied to much longer axle motif by polymerizing corresponding origami monomers. Successful formations of the complexes are confirmed by TEM, super-resolution fluorescent microscope, and by high-speed AFM.

TEM images are all beautiful and the complex structures are convincing considering their thorough confirmation using various fine techniques. I recommend publication of this paper.

Reviewer #2 (Remarks to the Author):

Review on Nature Communications (NCOMMS-16-10117-T)

"Long-Range Movement of Large Mechanically Interlocked DNA Nanostructures" by List et al.

This manuscript describes a novel approach to generate nano-mechanical devices with relatively long-range motions (10 up to several 100 nm) using DNA origami based nanostructures. The rotaxanes consist of an axle module with bulged stopper ends, and a ring module that can be opened and closed. Two alternative ways were used to generate the rotaxanes, both based on different designs of the DNA origami that each module uses one separate scaffold strand. The axle module is attached to the inner side of the ring while the ring is open. Next, the ring is closed by DNA hybridization. The yields of the assemblies are high (>85% without purification). Finally the attachment points between the axle and the ring are released by strand displacement to free the ring module so that it slides along the axle module, simply powered by random thermal motion. The axle module was also extended by polymerize multiple subunits to generate a long sliding axle (micrometer long). The relative motion of the two modules before and after the release was observed by AFM, TEM, bulk FRET measurements, super-resolution fluorescence imaging enabled by DNA-PAINT and also in real time by fast-scan AFM.

The designs have combined different approaches developed in the field of structural DNA nanotechnology, including creating rigid solid blocks of parallel helices, and of bended helices, creating larger structures using multiple subunits, making crossovers between parallel helices and with 90 degree angles, strand displacement using release strands and anti-release strands. FRET (dye labeling) and nanoparticle labeling aids in the assays to test the relative axle movement and rotations around the axles. The manuscript is well written, quality of the data is overall good and convincing. I recommend publish the manuscript in Nature Communications.

I have found the authors addressed all of my minor issues raised in the review to the submission to Nature Nanotechnology.

Reviewer #3 (Remarks to the Author):

This is an impressive paper describing the assembly of an origami-based rotaxane nanostructure that reveals long-range movement on a large mechanically interlocked structure. In contrast to previous DNA rotaxane structures that revealed structural flexibility and substantially smaller "moving" domains, the present origami structure allows the motility of the interlocked ring along longer scales.

The study is well performed, and the structure and function of the systems are supported by experiments. The structure of the origami-rotaxane is imaged by cryo-TEM experiments, and the motility of the interlocked ring is followed by fluorescence and single-molecule time-dependent fluorescence imaging.

The publication of the paper in Nature Communications is fully supported.

One minor comment:

The following references should be added to the paper:

1. The paper, Nano Lett., 13, 6275-6280 (2013), describing a rigid DNA rotaxane stoppered by Au nanoparticles, should be added.
2. The recent JACS Perspective, J. Am. Chem. Soc., 138, 5172-5185 (2016), on interlocked DNA nanostructures, should be added to the list of references.

Reply to the reviewer's requests:

We thank all reviewers for their extremely positive assessment of our revised manuscript. Reviewers #1 and #2 recommended publication as is.

Reviewer #3 had "One minor comment: The following references should be added to the paper:

1. The paper, Nano Lett., 13, 6275-6280 (2013), describing a rigid DNA rotaxane stoppered by Au nanoparticles, should be added.
2. The recent JACS Perspective, J. Am. Chem. Soc., 138, 5172-5185 (2016), on interlocked DNA nanostructures, should be added to the list of references."

We added these papers to the references as requested.